# Germline-encoded specificities and the predictability of the B cell response

**Marcos C. Vieira**[1]*, **Anna-Karin E. Palm**[2], **Christopher T. Stamper**[3,4], **Micah E. Tepora**[2], **Khoa D. Nguyen**[5], **Tho D. Pham**[5], **Scott D. Boyd**[5], **Patrick C. Wilson**[2,6], **Sarah Cobey**[1]*

1 Department of Ecology and Evolution, University of Chicago, Chicago, United States of America, 2 Department of Medicine, Section of Rheumatology, University of Chicago, Chicago, United States of America, 3 Center for Infectious Medicine, Department of Medicine Huddinge, Karolinska Institutet, Karolinska University Hospital Huddinge, Stockholm, Sweden, 4 Committee on Immunology, University of Chicago, Chicago, United States of America, 5 Department of Pathology, Stanford University School of Medicine, Stanford, United States of America, 6 Gale and Ira Drukier Institute for Children's Health, Weill Cornell Medicine, New York City, United States of America

* mvieira@uchicago.edu (MCV); cobey@uchicago.edu (SC)

**Data Availability Statement:** Code for the analyses is available at http://github.com/cobeylab/v_gene_selection. Data, intermediate files, and results are available on Zenodo (https://doi.org/10.5281/

## Abstract

Antibodies result from the competition of B cell lineages evolving under selection for improved antigen recognition, a process known as affinity maturation. High-affinity antibodies to pathogens such as HIV, influenza, and SARS-CoV-2 are frequently reported to arise from B cells whose receptors, the precursors to antibodies, are encoded by particular immunoglobulin alleles. This raises the possibility that the presence of particular germline alleles in the B cell repertoire is a major determinant of the quality of the antibody response. Alternatively, initial differences in germline alleles' propensities to form high-affinity receptors might be overcome by chance events during affinity maturation. We first investigate these scenarios in simulations: when germline-encoded fitness differences are large relative to the rate and effect size variation of somatic mutations, the same germline alleles persistently dominate the response of different individuals. In contrast, if germline-encoded advantages can be easily overcome by subsequent mutations, allele usage becomes increasingly divergent over time, a pattern we then observe in mice experimentally infected with influenza virus. We investigated whether affinity maturation might nonetheless strongly select for particular amino acid motifs across diverse genetic backgrounds, but we found no evidence of convergence to similar CDR3 sequences or amino acid substitutions. These results suggest that although germline-encoded specificities can lead to similar immune responses between individuals, diverse evolutionary routes to high affinity limit the genetic predictability of responses to infection and vaccination.

## Author summary

Antibodies arise as B cell receptors encoded by the stochastic recombination of immunoglobulin genes. While those genes evolve over millions of years, the receptors themselves evolve within weeks as B cells compete under selection for improved antigen recognition. This competition shapes the response to infection and vaccination; how

zenodo.8239755) and fastq files are available on SRA (PRJNA1005266).

**Funding:** This project has been funded in part with Federal funds from the National Institute of Allergy and Infectious Diseases, National Institutes of Health, Department of Health and Human Services under grant DP2 AI117921, CEIRS Contract No. HHSN272201400005C and CEIRR contract 75N93021C00015 to S.C., grants R01AI127877, R01AI130398, U19AI057229 to S.D.B and grants U19AI082724, U19AI109946, U19AI057266, CEIRS contract HHSN272201400005C, CEIRR contract 75N93019R00028 and CIVIC contract 75N93019C00051 to P.C.W. The content is solely the responsibility of the authors and does not necessarily represent the official views of the NIAID or the National Institutes of Health. The project was also partly supported by a Complex Systems Scholar Award from the James S. McDonnell Foundation awarded to S.C. M.C.V. was partly supported by a William Rainey Harper Dissertation Fellowship awarded by the University of Chicago. The funders had no role in study design, data collection and analysis, decision to publish, or preparation of the manuscript.

**Competing interests:** C.T.S. has consulted for Alvea / Telis Bioscience Inc. on the design of universal influenza vaccines. The other authors report no competing interests.

much the outcome depends on the initial choice of immunoglobulin genes versus subsequent receptor evolution is an open question that informs the predictability of the immune response and the long-term evolution of immunoglobulins. In simulations, we show that immunoglobulin genes with hardcoded specificity for the antigen can lead to either transient or persistent similarity in the response of different individuals. When the initial advantage is large relative to the effects of mutation, B cells using the same genes consistently dominate the response across individuals. Weaker initial advantages lead to similar responses early on but are later overcome by B cell evolution playing out differently in each individual due to chance events. We observe such increasingly divergent responses in mice infected with influenza virus. While long-term selection might hardcode specificities for particular pathogens on immunoglobulin genes, our results suggest diverse paths to potent antibodies can nonetheless limit the predictability of the response.

## Introduction

Antibodies owe their diversity and potency to evolution on two timescales. B cell receptors, the precursors of secreted antibodies, are encoded by immunoglobulin genes that have diversified over hundreds of millions of years [1–3]. Within individuals' lifetimes, these genes recombine to produce B cells with unique receptors [4–6]. The result is a diverse repertoire of naive (antigen-inexperienced) B cells collectively capable of binding virtually any antigen. Once activated, naive B cells expand into lineages that compete with each other for access to antigen. These lineages can acquire somatic mutations that improve binding, a process known as affinity maturation [7–9]. The sizes of the lineages, the sites they target, and their affinities for those sites shape the antibody repertoire's breadth and protectiveness.

Throughout this process, affinity for antigen—the biggest component of B cell fitness—is determined by the amino acid sequence of the B cell receptor. B cell fitness thus depends on the identities of the recombined germline immunoglobulin genes that encode the receptor and on subsequent somatic mutations. Affinity maturation can vastly improve binding [10, 11], yet high affinity for particular epitopes can be "hardcoded" on individual germline alleles from the start [12–19]. (These differences in the propensity to form high-affinity receptors may exist between germline genes occupying different loci or between different alleles occupying the same locus in different chromosomes. For simplicity, we will refer to "different alleles" in both cases.) For instance, some antibodies bind to epitopes via germline-encoded motifs in loops known as complementarity-determining regions (CDRs). CDRs 1 and 2 are encoded solely by the V (variable) allele, whereas CDR3 spans the junction of the V allele with joining (J) and, in the case of the heavy chain, diversity (D) alleles. Because they do not arise directly from recombination, CDRs 1 and 2 are presumably less affected by epistasis than CDR3 and suggest how some germline V alleles could be reliably selected to respond to particular antigens.

A germline-encoded specificity might have arisen as an evolutionary spandrel, a byproduct of immunoglobulin gene diversification in vertebrates, and then been selected [20–22]. For instance, germline alleles with innately high affinity for bacterial antigens might be favored if they protect against commonly encountered pathogens and commensals via broad classes of epitopes shared by these organisms [14, 16, 23].

How much the development of high antibody titers to an antigen depends on the availability of germline alleles already specific to that antigen, or with greater potential to adapt to that antigen, is unclear [22]. More broadly, it is unclear how many evolutionary paths and solutions there are to similar phenotypes, i.e., individuals with high antibody titers. The existence of few paths would appear as consistent selection of the same germline alleles into the response across individuals and selection for specific CDR3 sequences and somatic mutations during affinity maturation. Past studies conflict on how "convergent" antibody responses are. In some cases, only a few germline alleles are represented in the response (e.g., [24–28]), suggesting that germline-encoded specificities determine the outcome of B cell competition. In other cases, most germline alleles are used (e.g. [29–33]), only some of which appear significantly overrepresented, suggesting weaker selection for specific germline alleles and more solutions. Those studies vary not only in the complexity of the antigen and the type of B cell studied but also in the amount of time since the exposure and thus the extent of affinity maturation. Understanding the role of germline genetic variation in the response to an antigen might help explain individual variation in responses to infection and vaccination and inform the feasibility of some vaccination strategies.

Using computational models and experiments, we investigated the role of germline-encoded specificities during the B cell response. In simulations, whether germline-encoded advantages lead to similar allele usage in different individuals depends on how large they are relative to the rate and effect size variation of somatic mutations. While large germline-encoded advantages result in persistent similarity between individuals throughout the response, affinity maturation can overcome weaker initial differences and lead to increasingly divergent responses over time due to the stochasticity inherent to evolution. We observe this latter pattern in mice experimentally infected with influenza—a virus that does not naturally infect them—suggesting that germline-encoded specificities that arise as byproducts of evolution are initially weak and thus mostly affect the early B cell response. Selection to reinforce those specificities in the long-term evolution of jawed vertebrates might thus be driven by the fitness benefits of responding rapidly to common pathogens. The lack of consistency in germline allele usage or specific mutations later in the mouse response suggests no pronounced differences in the adaptability of different immunoglobulin genes or naive B cell repertoires to a complex antigen such as influenza.

## Results

To quantify how strongly germline-encoded advantages shape the B cell repertoire, we use the ratio of a germline allele's frequency in the population of activated (responding) B cells to its frequency in the naive repertoire. Within a single individual, this ratio reflects the overall activation and growth of B cell lineages using a particular germline allele relative to lineages using other alleles. If lineages using a particular allele have consistently higher initial affinity (or if they are more likely than others to evolve high affinity via affinity maturation), the experienced-to-naive ratio for that allele should be consistently greater than 1 in different individuals (Fig 1, top). In contrast, if growth rates are poorly predicted by the use of individual germline alleles and depend more strongly on chance events during recombination and subsequent evolution, which germline alleles successful lineages use will often vary between individuals (Fig 1, bottom). Thus, we hypothesize that a strong correlation in experienced-to-naive ratios between individuals indicates that germline-encoded differences shape the outcome of B cell evolution, whereas weak correlations would indicate the dominance of other factors.

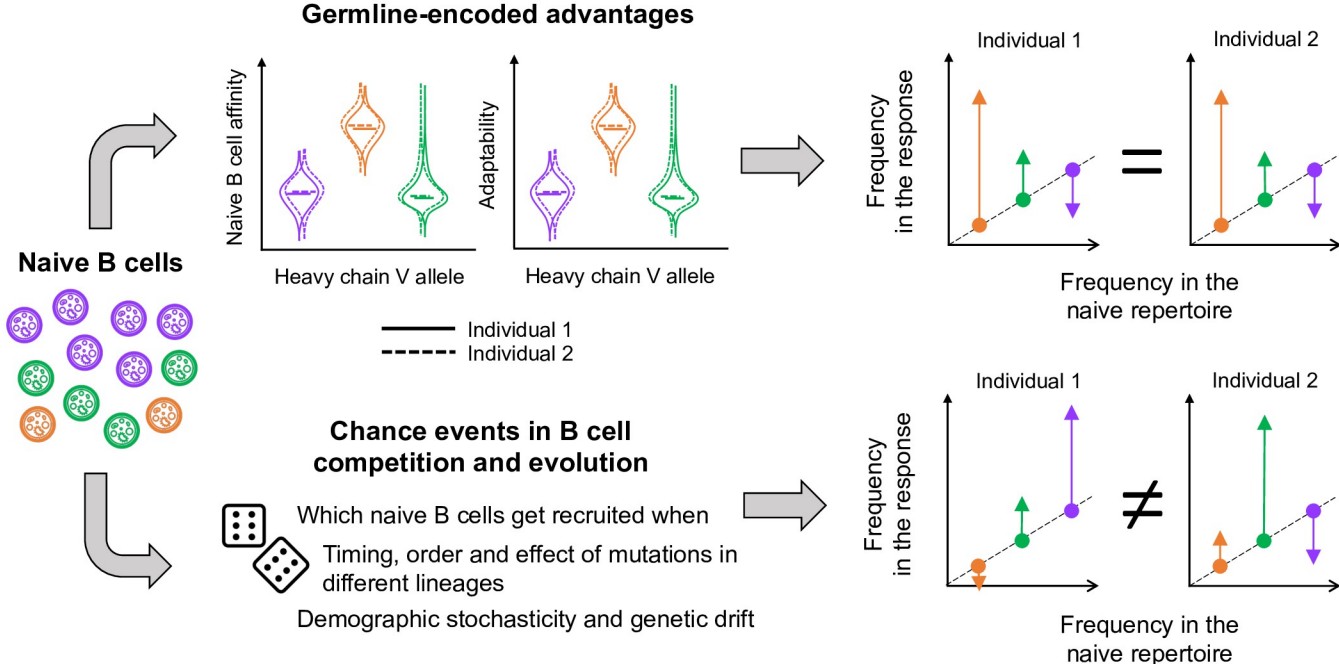

**Fig 1. Schematic of factors controlling germline allele frequencies in the B cell response to a particular antigen.** Three heavy-chain V alleles (orange, purple and green) are present at different frequencies in naive B cells. Although they have the same heavy-chain V allele, naive cells of the same color can have different alleles from the other sets in the heavy and light chains (and different insertions and deletions at the alleles' junctions). Different combinations produce receptors with different affinities for the antigen and different propensities for adaptation during affinity maturation. If these distributions vary between heavy-chain V alleles, alleles more likely to produce receptors with high affinity or high adaptability will tend to increase in frequency relative to the naive repertoire. These deviations are expected to be consistent in individuals sharing similar sets of germline alleles at similar frequencies in the naive repertoire. However, which B cell lineages dominate the response—and what heavy-chain V alleles they happen to use—is also contingent on events that are largely unpredictable, potentially leading to uncorrelated frequency deviations in the response of different individuals.

We describe these ideas in more detail below and then evaluate them in simulations and experiments.

## Why fitness distributions might vary between germline immunoglobulin alleles, and how chance events might trump those differences

Fitness differences between B cells using different germline alleles might arise from differences in initial affinity, in the propensity to adapt during affinity maturation, or both.

*Germline-encoded affinity*. Since affinity is a property of the entire receptor, not of its individual constituent alleles, naive B cells using a particular germline allele have a distribution of possible affinities determined by stochastic recombination with other alleles and insertions and deletions at their junctions (the breadth of this distribution reflects the strength of epistasis between germline alleles). Within an individual, different germline alleles will likely have different affinity distributions. Naive B cells using a specific heavy-chain V allele might bind an antigen well across all combinations with other alleles via CDRs 1 and 2 (Fig 1, orange allele), whereas naive B cells using another V allele might bind uniformly poorly (Fig 1, purple allele) or have high affinity only in certain combinations (Fig 1, green allele). The affinity distribution of an allele may vary between individuals, since individuals vary in their germline immunoglobulin diversity and expression [34–36].

*Germline-encoded adaptability.* The potential for a B cell receptor to evolve higher affinity is also a property of the entire receptor, not of individual alleles. Receptors using different alleles might have different propensities to adapt (Fig 1), for instance, if they tend to have different rates of beneficial and deleterious mutations. Variation in the expected impact of mutations occurs not only from differences in structure that might affect mutational tolerance but also because the enzymes responsible for mutating the B cell receptor target different nucleotide motifs at different rates [37–40]. Given a mutation, the relative probabilities of beneficial and deleterious impacts fundamentally arise from epistasis: mutations are more likely to change affinity or disrupt function in some backgrounds than in others [41, 42].

The realized growth rates of B cell lineages might deviate from these germline-encoded expectations due to several sources of stochasticity:

*Stochasticity in B cell activation and in the colonization of germinal centers.* Because the number of naive B cells is finite and the probabilities of different VDJ recombinations vary by orders of magnitude [43], rare germline allele combinations with high affinity may be present in the naive repertoires of some individuals but not others. Even if present in most individuals, low-frequency, high-affinity germline-allele combinations might by chance be recruited only in some of them. Additionally, lineages that happen to arrive first in germinal centers might prevent others with higher initial affinity from establishing later, but the strength of "priority effects" in affinity maturing B cells is unclear [44, 45].

*Stochasticity in the timing, order, and effect of mutations.* Which lineages ultimately evolve the highest affinity and outcompete others depends on the timing, order, and effect of mutations in each lineage. This concept is part of clonal interference dynamics [46]. Epistasis can increase the impact of mutation order [47].

*Demographic stochasticity and genetic drift.* Demographic stochasticity and genetic drift might be especially important early in the response when population sizes are small. Some lineages might go extinct purely by chance, and genetic drift might cause new mutations to fix within a lineage even if they are neutral or deleterious. The loss of newly arisen beneficial mutations due to drift is important even in large populations [48].

## Large germline-encoded advantages lead to persistent similarity between individuals, while smaller ones are overcome differently in each individual

To evaluate when germline-encoded fitness differences might be overwhelmed by chance events during recombination and affinity maturation, we used a stochastic mathematical model to simulate B cell evolution and competition in germinal centers (Methods: "Model of B cell dynamics"; Table 1). We focused on how similarity between individuals over time depends on the magnitude of germline-encoded advantages (*s*, defined as the increase in mean naive affinity for an allele relative to other alleles) and the rate and effect size variation of somatic mutations (with variation measured as the standard deviation $\beta$ around a mean effect of zero). Different host-pathogen systems likely exist in different parts of this parameter space, since naive affinities and mutational effects are epitope-specific and depend on the host's germline alleles, their recombination and insertion/deletion probabilities, and the frequency and targeting of somatic hypermutation. The model focuses on the subset of the B cell response derived from germinal centers without considering extrafollicular B cell populations that expand outside of germinal centers, although reports of selection and somatic hypermutation in those populations suggest they have similar dynamics [29, 49]. To simulate selection for affinity, B

**Table 1. Default parameter values used in simulations.**

| Parameter | Symbol | Value |
|---|---|---|
| Baseline average naive B cell affinity | $a$ | 1 |
| Baseline standard deviation of naive B cell affinity | $\sigma$ | 1 |
| Expected number of lineages seeding each GC | $I_{\text{total}}$ | 200 |
| GC carrying capacity | $K$ | 2000 |
| Duration of GC immigration phase | $t_{\text{imm}}$ | 6 days |
| Maximum rate of cell division | $\mu_{\text{max}}$ | 3 cell$^{-1}$day$^{-1}$ |
| Death rate | $\delta$ | 0.2 cell$^{-1}$day$^{-1}$ |
| Standard deviation of mutation effect size (relative to $\sigma$) | $\beta$ | variable |
| Average affinity increment for high-affinity alleles (relative to $\sigma$) | $s$ | variable |
| Mutation rate of high-mutation alleles relative to baseline mutation rate | $\gamma$ | variable |

cells are stochastically sampled to immigrate or divide based on their affinity relative to other cells in the naive repertoire (in the case of immigration) or in the germinal center (in the case of division). Dividing B cells then undergo affinity-changing mutations with some probability. To represent variation in germline-encoded affinity, B cells using different heavy chain germline V alleles can have different naive affinity distributions. The variation within each distribution in turn represents the effects of stochasticity in VDJ recombination and the pairing of heavy and light chains. The model also allows different germline V alleles to have different mutation rates, representing one aspect of variation in adaptability. We simulated 100 individuals, varying the number of germinal centers per individual and assuming no migration occurs between germinal centers (increasing the number of individuals to 1000 produced no substantial changes; S1 Fig). For each simulated individual, we randomly sampled (with replacement) a mouse for which we empirically estimated the set of heavy chain V alleles and their frequencies in the naive repertoire. These mice typically had about 75 heavy-chain V alleles (60–70 of which were typically shared between a pair of mice), and allele frequencies in the naive repertoire were positively correlated between mice (Fig 4A-B). To simulate alleles with higher naive affinity or higher mutation rate than the others, we randomly sampled a set of 5 alleles present in all mice with average naive frequency of 1–2% (the typical median frequency in the naive repertoire) and used that same set of alleles across all mice.

We first considered the case in which all germline alleles have the same naive affinity distribution and the same mutation rate. In this "functional equivalence" scenario, individuals have similar allele frequencies early in the response, as germline alleles are initially recruited in proportion to their frequency in the naive repertoire. However, similarity between individuals decreases over time due to stochasticity, with the extent of this decline depending on the number of germinal centers per individual and the somatic hypermutation rate (Fig 2). The higher the mutation rate, the faster each germinal center tends to become dominated by a few large lineages with high affinity (Fig 2A). Without differences in initial affinity or adaptability between the germline alleles, the probability that one of these lineages uses a particular allele is equal to the allele's frequency in the naive repertoire. The smaller the number of germinal centers, the more the set of "winner alleles" represented in the dominant lineages tends to vary between individuals, causing allele frequencies to diverge over time and experienced-to-naive ratios to remain uncorrelated (Fig 2B). In contrast, increasing the number of germinal centers per individual increasingly allows germline alleles to be represented in proportion to their frequencies in the naive repertoire, so allele frequencies remain correlated between individuals throughout the response.

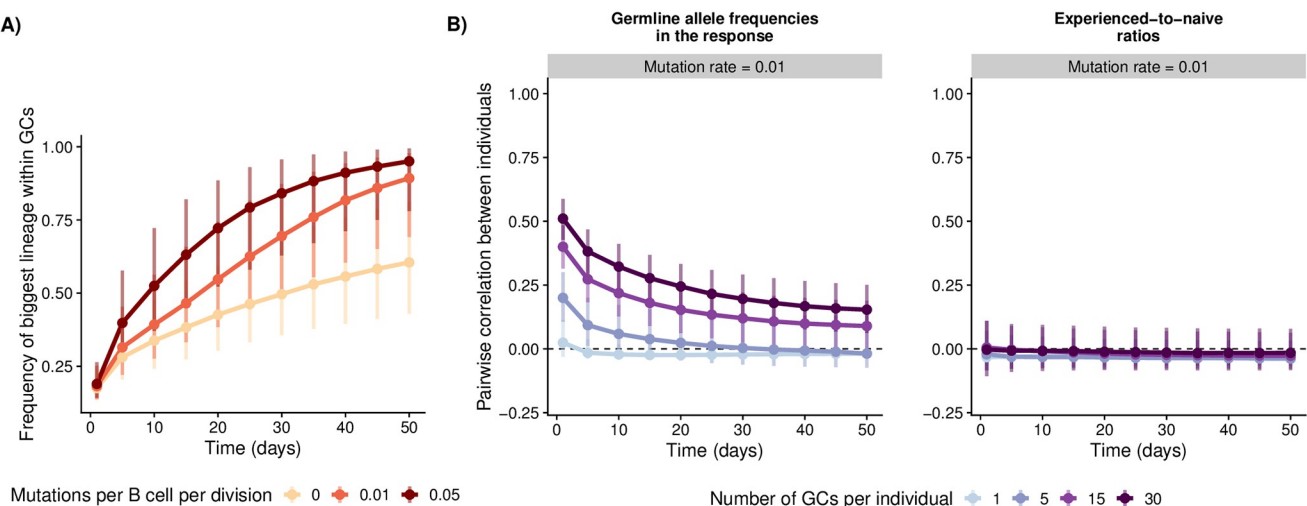

**Fig 2. Simulations in the scenario with identical naive affinity distributions and mutation rates across germline alleles.** For each parameter combination, we simulated 100 individuals with varying numbers of germinal centers. A) For each germinal center, we computed the fraction of the total germinal center population occupied by the biggest lineage. Points and vertical bars represent the median and the 1st and 4th quartiles for this fraction across simulated germinal centers. B) We measured germline allele frequencies across germinal centers in each individual and computed the ratio between these frequencies and the allele frequencies in the naive repertoire. We then computed the correlation in frequencies and experienced-to-naive-ratios between all pairs of individuals. Points and vertical bars represent the median and the 1st and 4th quartiles of these correlations across all pairs (i.e., the bars represent true variation in simulated outcomes and not the uncertainty in the estimate of the median). We set $\beta = 2$ and left other parameters as shown in [Table 1].

If some germline alleles tend to encode receptors with higher affinity than others, not only allele frequencies (S2 Fig) but also their deviations from the naive repertoire (Fig 3) are correlated between individuals early in the response. Whether this similarity between individuals persists over time depends on the rate and effect size of somatic mutations relative to the germline-encoded advantage of the high-affinity alleles. If the rate and effect size variation of mutations is small relative to that initial advantage, germinal centers are consistently dominated by B cell lineages using high-affinity alleles (S3 Fig), resulting in similar allele frequencies and experienced-to-naive ratios between individuals throughout the response. As the rate or the effect size variation of somatic hypermutation increases, so does the opportunity for B cell lineages using low-affinity alleles to overcome their initial disadvantage and reach high frequencies within germinal centers (S3 Fig). In this scenario, precisely which low-affinity alleles are used by lineages that do so is highly stochastic and varies between individuals, since all B cells are assumed to have the same probability of acquiring beneficial mutations irrespective of the germline V allele they use. As a result, the correlation in allele frequencies and the correlation in experienced-to-naive-ratios both decrease over time. We observed the same general behavior if we changed the number of high-affinity alleles to 1 or 10. (S4 Fig).

Finally, when some germline alleles have a higher mutation rate than others (and the baseline rate and effect size variation of mutations are sufficiently large), the faster mutating B cell lineages eventually dominate germinal centers due to their propensity to adapt (S5 Fig), countering the tendency for allele frequencies to become less correlated over time observed in the functional equivalence scenario and leading to a positive correlation in experienced-to-naive ratios later in the response (S6 Fig).

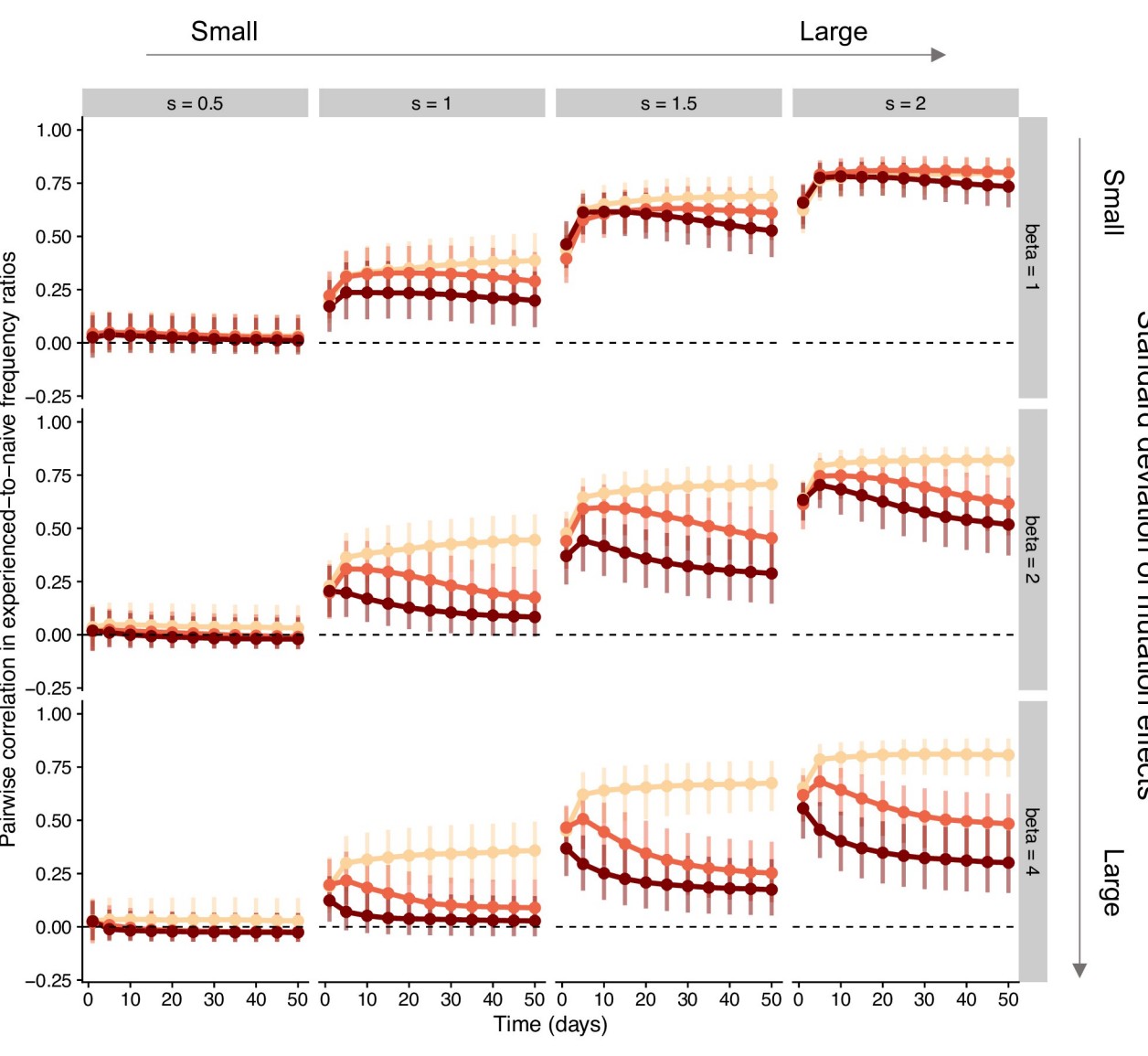

**Fig 3. Simulations in the scenario where the same 5 germline alleles have higher expected affinity than the others in all individuals.** For each parameter combination, we simulated 100 individuals with varying numbers of germinal centers. We measured germline allele frequencies across germinal centers in each individual and computed the ratio between these frequencies and the allele frequencies in the naive repertoire. We then computed the correlation in these experienced-to-naive-ratios between all pairs of individuals. Points and vertical bars represent the median and the 1st and 4th quartiles of these correlations across all pairs (i.e., the bars represent true variation in simulated outcomes and not the uncertainty in the estimate of the median). For these simulations, we assumed 15 germinal centers per individual. Other parameter values are as in Table 1.

## Decreasing similarity in the mouse response to influenza virus infection suggests affinity maturation overcomes germline-encoded differences

We compared these simulated dynamics with the B cell response of C57BL/6 mice infected with influenza virus once or twice and sacrificed at different times points (8, 16, 24, 40 and 56 days after the primary infection, with the secondary infection on day 32; Materials and methods: "Experimental infection of mice with an influenza A/H1N1 virus"). Because influenza

viruses do not naturally infect mice, any germline-encoded specificities for influenza antigens are either evolutionary spandrels or the product of selection to recognize molecular patterns shared between influenza and pathogens that have historically infected mice. We used RNA sequencing of the heavy chain to estimate the frequencies of germline alleles and the relative sizes of B cell lineages in each mouse. While we did not sort cells based on the ability to bind influenza, we found that uninfected controls had very few germinal center, plasma or memory cells in the mediastinal lymph node (S7 and S8 Figs), suggesting the lymph-node B cells of infected mice were induced by influenza in agreement with previous work [50]. We therefore focused on lymph node B cells, with potentially many different specificities for influenza antigens. Early in the response, lymph node populations likely consist of extrafollicular plasma cells expanding outside of germinal centers, with germinal-center derived cells arriving later [50] and persisting for as long as six months [51].

Although C57BL/6 mice are inbred, we found differences in the set of germline V alleles between mice that could not be explained by variation in sequencing depth and that were robust to the choice of reference allele set and annotation tool (S9 Fig; Methods: "Estimating the frequencies of V alleles and B cell lineages"). To our knowledge, a systematic analysis of germline variation between mice of the same inbred strain under different reference sets and annotation tools is missing, though at least one other study suggests some variation in C57BL/6 [52]. Despite the variation we observed, mice had similar sets of germline alleles present at correlated frequencies in the naive repertoire (Fig 4). Because this correlation was nonetheless weaker than previously reported by [52], we repeated the analysis using data from that study to compute germline allele frequencies in the naive repertoire.

As expected, influenza infection led to affinity maturation in mouse B cell lineages (S10 Fig). Serum antibody titers against the infecting virus measured by ELISA rose approximately 1,000-fold between days 8 and 24 and remained high. In parallel to this rise in antibody titers, germinal center and plasma cell populations became increasingly dominated by a few lineages, suggesting that lineages varied in fitness due initial differences in affinity, differences acquired during the lineages' subsequent evolution, or both. Lineages sampled at later time points had more high-frequency amino acid mutations within them (those present in 50% or more of the reads in a lineage). Those mutations include fixed mutations and those potentially rising to fixation via selection for affinity, and they are unlikely to be deleterious or to have arisen from sequencing and amplification errors (which we estimate at 1.8 per thousand nucleotide bases; Materials and methods: "B cell receptor sequencing"). These trends were visible in the lymph nodes of infected mice but not apparent in other tissues or in control mice (S11 Fig), suggesting they were driven by the influenza infection. (Influenza-specific lineages may have been present in other tissues, but our data do not allow us to distinguish them from lineages elicited by other antigens.)

Most germline V alleles observed in a mouse (across all tissues and cell types sampled) were represented in the influenza-induced lymph node populations (Fig 4), suggesting that most mouse V alleles can produce at least some receptors capable of binding influenza antigens. We measured the correlation in germline V allele frequencies and experienced-to-naive ratios between pairs of infected mice and compared the observed patterns with a null model in which a lineage's fitness is independent of which germline V allele it uses, mimicking the equivalent-alleles scenario in our simulations. We did so by keeping the observed distribution of lineage sizes (a proxy for lineage fitness) while randomly assigning each lineage's germline V allele based on naive repertoire frequencies.

In plasma cells and germinal center cells, allele usage became increasingly dissimilar between mice over time despite evidence of germline-encoded advantages, suggesting those advantages were overcome by divergent B cell evolution in each mouse. In both cell types, the

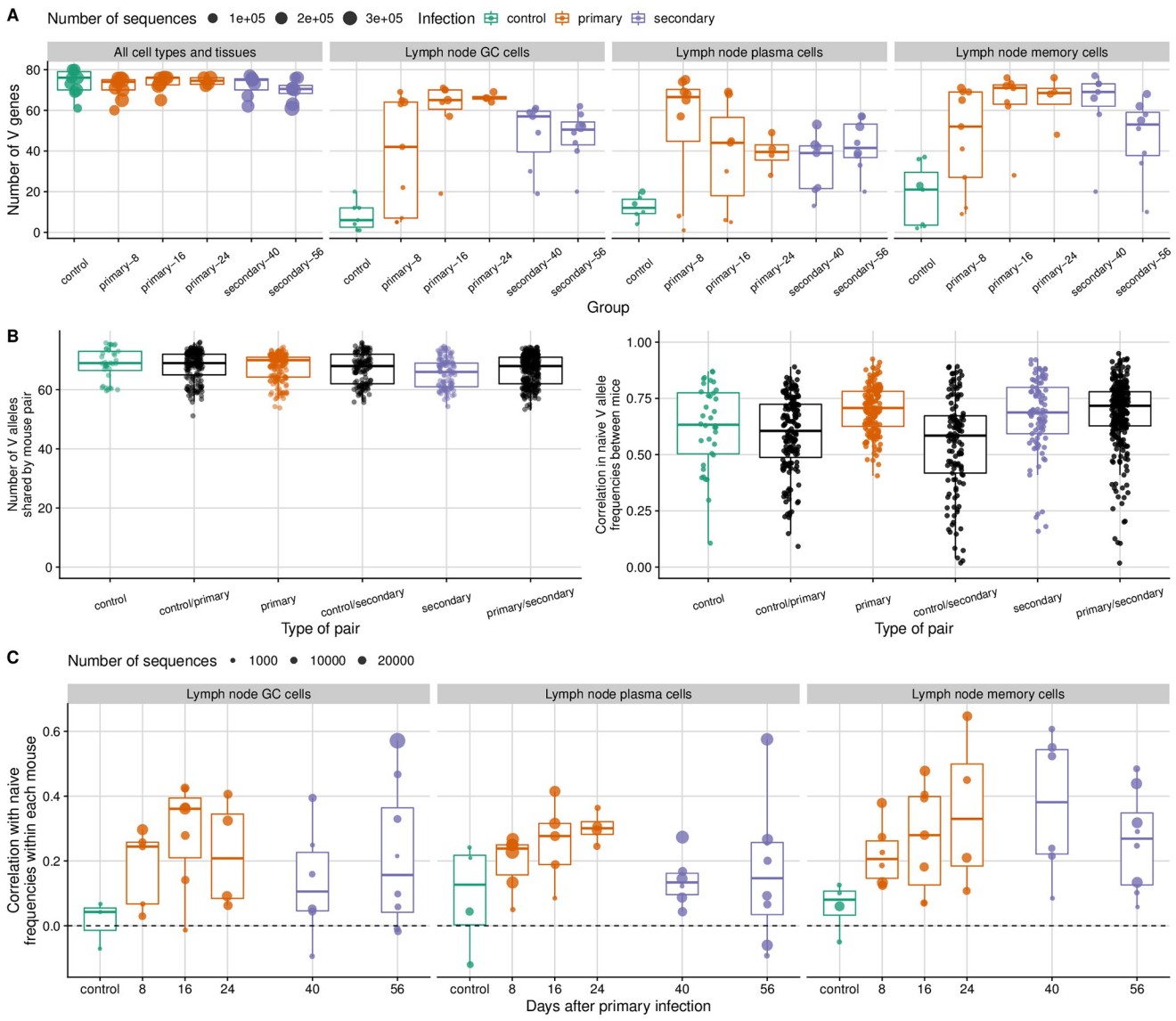

**Fig 4. Immunoglobulin V gene usage in the mouse B cell response to influenza infection.** (**A**) The number of germline immunoglobulin V alleles is shown for mice infected once or twice with a mouse-adapted H1N1 virus and sacrificed at different time points (8, 16, 24, 40 and 56 days after the primary infection, with the second infection at day 32). Uninfected control mice are shown in green. Each point represents a mouse. At the peak of the response, most alleles present in each mouse are represented in lymph-node germinal center (GC), plasma and memory cells, which were likely induced by the influenza infection. (**B**) Number of V alleles shared by pairs of mice in the naive repertoire (left) and the Pearson correlation in their frequencies for each pair (excluding mice with fewer than 100 reads in the naive repertoire; right). Each point represents a pair. (**C**) Pearson correlation within each mouse between V allele frequencies in influenza-induced populations and frequencies in the naive repertoire. Each point represents a mouse, and solid-line boxplots indicate the distribution in the observed data.

correlation in allele frequencies between mice was stronger than expected under the null model early in the response but close to the null expectation later on (Fig 5A, left panel). Early on in plasma cells, and throughout the response in germinal center cells, some germline V alleles were consistently overrepresented in different mice, leading to correlated experienced-to-naive ratios (Fig 5A, right panel; S12 and S13 Figs) and suggesting those alleles contributed to higher affinity or adaptability than did others. For instance, in day-8 plasma cells, IGHV14-

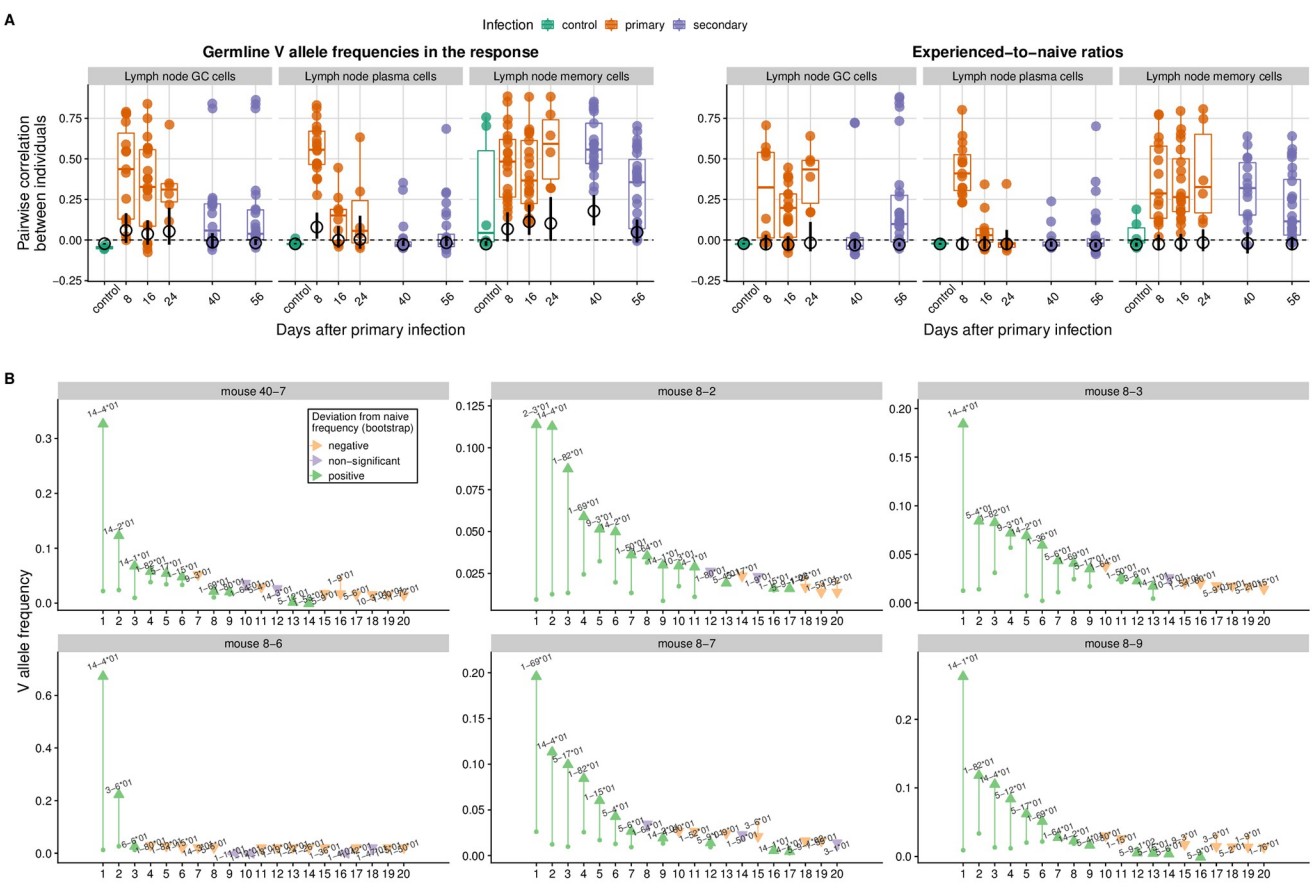

**Fig 5. Correlation between mice in the V allele frequencies of influenza-induced populations and in the deviations of those frequencies from the naive repertoire.** (**A**) Distribution of pairwise correlations at each time point. Each colored point represents a pair of mice with at least 100 reads each in the respective B cell population. The horizontal bars indicate the observed median across mouse pairs, whereas the black circles and black vertical bars indicate the bootstrap average and 95% confidence interval for the median in a null model with V alleles randomly assigned to each B cell lineage ($n = 500$ randomizations). We computed correlations using Pearson's coefficient and measured frequency deviations as the ratio between a V allele's frequency in an influenza-induced population and its frequency in the naive repertoire. (**B**) Frequency of the 20 most common V alleles in the lymph node plasma cells of each mouse 8 days after primary infection. Each panel represents an individual mouse. The arrows go from each allele's frequency in the naive repertoire to its frequency in lymph node plasma cells. Each allele was labelled as significantly over- or underrepresented in each mouse if the ratio of its experienced and naive frequencies was outside a 95% confidence interval obtained by bootstrap sampling ($n = 500$) of experienced frequencies from the naive repertoire (preserving the observed total number of sequences in each mouse). Mouse 40-7, which was sacrificed 8 days after the secondary infection, was considered a day-8 primary-infection mouse because it showed no signs of infection after the first inoculation and had ELISA titers similar to those of day-8 infected mice.

4*01 increased in frequency relative to the naive repertoire in all 6 mice with enough data, becoming the most common V allele in 3 mice and either the second or the third most common in the other 3 (Fig 5B). Later in the plasma cell response, however, experienced-to-naive ratios became uncorrelated between mice: the most common V allele was usually different in different mice, and most germline alleles were overrepresented relative to the naive repertoire in some mice but not in others (S12 Fig). Taken together, these results suggest that some mouse V alleles are more likely than others to make good receptors against influenza, but that those germline-encoded advantages are not strong enough to drive persistently similar responses given the role of chance in B cell evolution.

In contrast with plasma and germinal center cells, memory cells showed persistently biased V gene usage in influenza-infected mice (Fig 5A and S14 Fig). Since activated B cells with low

affinity are more likely than others to exit germinal centers and differentiate into memory cells [53], dominant lineages with high affinity for influenza antigens might contribute less to the memory cell population than they do to the germinal center and plasma cell populations. Consistent with that possibility, the increasing dominance by a few large and mutated lineages seen in germinal center and plasma cells of infected mice was not evident in their memory cells (S10 Fig). Thus, memory cells at different times during the response might represent a recent sample from the naive repertoire, reflecting germline-encoded differences in initial affinity and less the divergent outcomes of affinity maturation.

We found similar results using a different germline annotation tool and different germline reference sets, using independent sequence data from [52] to estimate naive allele frequencies, and when computing allele frequencies based only on unique sequence reads (collapsing identical reads from the same mouse, tissue, cell type and isotype) (S16 Fig).

## Germline V alleles consistently overrepresented early in the response have low predicted mutability in CDRs

Our simulations suggest that correlated experienced-to-naive ratios early in the response more likely reflect germline-encoded differences in affinity than in the overall mutation rate (Fig 3 and S6 Fig; we did not consider that some germline alleles may have relatively more beneficial mutations accessible to them, which might also lead to germline-encoded differences in adaptability). With sequence data alone, we cannot determine if germline V alleles overrepresented in the early mouse response generate receptors with especially high affinity for influenza antigens. We can, however, estimate potential differences in the overall mutation rate between germline alleles based on their sequences alone, using estimates of the propensity of different nucleotide motifs to undergo somatic hypermutation (although those estimates were derived from mouse light-chain rather than heavy-chain genes [54]) (Methods: "Mutability analysis of germline alleles").

Germline alleles with higher predicted mutability in the CDRs, which might give those alleles a higher rate of affinity-enhancing mutations, did not have a clear advantage (S17 Fig). Neither did germline alleles with lower mutability in the structurally-important framework regions (FRs; where mutations are more likely to be deleterious than in CDRs). Instead, in day-8 plasma cells we found the opposite relationship: germline alleles tended to increase in frequency relative to the naive repertoire if they had high mutability in FRs and low mutability in CDRs. The consistently overrepresented and dominant allele IGHV14-4*01, for instance, is predicted to be one of the least mutable in CDRs 1 and 2, (S17 Fig). Other germline alleles consistently overrepresented in day-8 plasma cells, IGHV1-82*01, IGHV1-69*01, IGHV14-1*01 and IGHV14-2*01 (5 of 6 mice with enough data) have similarly low predicted mutability in CDR2, though not in CDR1. If those alleles do have a high propensity to bind influenza antigens, low mutability in CDRs 1 and 2 might reduce the chance that mutations disrupt this initial binding, potentially reinforcing the fitness advantage of B cells using those alleles.

## B cell lineages sharing the same germline V allele rarely had mutations in common

While germinal center cells and plasma cells were increasingly dominated by large lineages with somatic mutations, the sheer number of high-frequency mutations acquired by a B cell lineage did not predict its success (S18 Fig). This observation is consistent with previous work showing that the number of mutations in the B cell receptor does not predict affinity or neutralization strength [33, 53, 55]. Thus, successful lineages might be those that acquire few substitutions with large effects on affinity, instead of many substitutions with smaller effects.

Most pairs of lineages with the same V allele had no high-frequency mutations in common (S19 Fig). For specific cell types and specific V alleles, we found some instances of high-frequency mutations shared by multiple lineages. However, they were constrained to one or two mice, suggesting they might be an artifact of the incorrect partitioning of a single large lineage into several small ones. Independent acquisition of the same mutation has been shown in lineages using the same heavy chain V allele and recognizing the same antigen or epitope [13, 56, 57]. Our analysis differs from those previous observations by investigating convergent mutations across V alleles in a population with multiple potential specificities, suggesting that convergent mutations may be rare overall when considering a whole pathogen.

### Influenza antigens do not strongly select for specific CDR3 sequences

While binding can occur via the two CDRs solely encoded by the V segment, it often occurs via CDR3, which spans the junction between the segments. Thus, influenza antigens might select for receptors with specific CDR3 sequences, despite showing little selection for specific germline alleles encoding CDR1 and CDR2. To investigate this possibility, we computed the amino acid sequence and biochemical similarity of CDR3 sequences sampled from different mice and matched for the same length (Methods: "Measuring CDR3 sequence similarity"). On average, length-matched CDR3 sequences from the influenza-induced populations of different mice were no more similar than sequences sampled from their naive repertoires (S20 Fig). (We found similar results when matching sequences both for length and heavy-chain V allele, suggesting the choice of V allele does not strongly constrain the sequence of CDR3.) While individuals exposed to the same pathogen often share receptors with specific CDR3 sequences [32, 57–59], our results suggest those convergent CDR3s do not make up a large fraction of the response.

Finally, we found no evidence that the germline alleles overrepresented in the early B cell response to influenza were associated with more diverse CDR3 sequences in the naive repertoire (S21 Fig).

### Discussion

How much germline-encoded specificities shape the B cell response is important for adaptive immune evolution and vaccination strategies. In simulations, we show that whether germline-encoded advantages lead to persistently similar allele usage between individuals depends on the magnitude of those advantages relative to the effect of mutations. When mutations can easily overcome initial differences in affinity, germline allele usage diverges after an initial period of similarity. This divergence happens because B cell evolution and competition depend on factors that are largely unpredictable, such as the timing, order and effect of mutations in different lineages, genetic drift and demographic stochasticity. In mice infected with influenza, these "historical contingencies" [60,61] erase what seem to be relatively small germline-encoded advantages that we expect to have evolved by chance, given that influenza does not naturally infect mice. As is often the case in evolution [62], the overall phenotype is remarkably robust to divergence in genotypes: potent antibodies emerge in most individuals, suggesting there are many different ways to achieve high affinity against the same pathogen.

The diversity of allele usage in a B cell response likely depends on the complexity of the antigen. Although certain amino acid motifs can make germline alleles polyreactive [63, 64], individual alleles are unlikely to have a consistent advantage over others across all antigens and epitopes in a pathogen [19]. The response to haptens (simple antigens with few potential epitopes) tends to be dominated by one or a few alleles [25, 26], whereas the response to complex antigens can use many [30]. Only a few germline alleles are represented in monoclonal

antibodies specific for narrowly defined sites on influenza hemagglutinin [27, 28], while tens of alleles encode monoclonal antibodies that bind epitopes in the major domains of the SARS-CoV-2 spike protein [32, 33]. Many germline V alleles are used by monoclonal antibodies against the IsdB protein of *Staphylococcus aureus*, but antibodies targeting each epitope tend to use only one or two of them [14].

If responses to each epitope are consistently dominated by one or a few alleles, similarity in allele usage in the overall response might reflect the extent to which different individuals target the same epitopes. In mice infected with influenza, divergent allele usage over time suggests individuals target increasingly divergent sets of epitopes. Consistent with this possibility, mouse responses to infection with another H1N1 strain initially focused on a single site on the hemagglutinin head but later targeted different sites more evenly, although it is unclear how much diversity in epitope targeting increased within versus between mice [65]. Yet the causal relationship between immunodominance—biases in the immune response toward specific antigens and epitopes [66]—and germline-encoded specificities is unclear. Physical attributes that render certain epitopes immunodominant (e.g., their accessibility) might also explain why germline alleles specific for those epitopes tend to encode receptors with high affinity. Alternatively, an epitope might dominate over others with comparable accessibility *because* a specific germline allele binds it better than other alleles bind their preferred epitopes [19]. Our simulations did not consider the dynamics of adaptation to different epitopes.

A testable prediction suggested by our results is that germline allele usage might diverge between people following repeated exposures, such as sequential influenza infections or vaccines, or over time during chronic infections, such as HIV. This prediction depends on the magnitude of germline-encoded advantages and on extent to which responses to repeated or prolonged infections rely on the reactivation of preexisting memory cells and their reentry into germinal centers [67–71]. Divergence in germline allele frequencies might be small if the response to each exposure is dominated by lineages newly recruited from the naive repertoire. In contrast, successive bouts of evolution by recalled B cell lineages might increase the chance that they overcome germline-encoded advantages, especially if those advantages are small. Vaccine strategies focused on the recruitment of specific alleles [72–75] might be hindered in their immediate goal by contingent patterns of allele usage in different people, especially if the strategy involves multiple immunizations or immunizations in people with extensive immune memory.

Further understanding variation in germline gene usage requires overcoming limitations of our analyses. Migration between germinal centers [76], which we did not investigate in our simulations, might lead to faster divergence, similar to the effect of having fewer germinal centers. We also did not consider germline-encoded differences in adaptability other than differences in the overall mutation rate. Although genetic drift, demographic stochasticity and priority effects in the colonization of germinal centers were modeled, we did not systematically explore their impacts. Understanding those processes might require fine-scale data on germinal center dynamics. Additional experiments could estimate the epitope-level specificity and affinity distribution of naive B cells using different germline alleles, compare variation within and between those distributions, and directly test if alleles with higher affinity distributions tend to be used by B cell lineages with high growth rates. Fine-scale specificity and affinity measurements could also help determine how much divergence in allele usage derives from differences in epitope targeting and whether certain germline alleles maintain persistent advantages with each epitope.

Finally, if germline-encoded specificities that arise by chance are weak and thus mostly affect the early B cell response, long-term selection to reinforce them might be linked to the benefits of responding rapidly to common and/or especially harmful pathogens. Mathematical

 

models suggest that maintaining innate defenses against a particular pathogen becomes more advantageous the more frequently the pathogen is encountered [77]. Germline alleles specific to common pathogens or pathogenic motifs might be selected, effectively hardcoding innate defenses into the adaptive immune system [23]. A reliable supply of receptors against common enemies might be especially important in small and short-lived organisms, which can more quickly die of infection and have fewer naive B cells with which to cover the vast space of possible pathogens [23]. Reinforcing germline-encoded specificities might also be especially useful when the opportunity for adaptation is limited, as might be the case for pathogens that induce extrafollicular responses without extensive B cell evolution (although affinity maturation can occur outside of germinal centers; [29, 49]). Understanding what conditions favor similar versus contingent allele usage in the antibody repertoire may thus shed light on the long-term evolution of immunoglobulin genes.

## Materials and methods

### Ethics statement

All mouse experiments were approved by The University of Chicago Institutional Animal Care and Use Committee (IACUC protocol 71981).

### Model of B cell dynamics

We modeled B cell evolution and competition in germinal centers using stochastic simulations based on a Gillespie algorithm. There are three types of independent events in the model: immigration of individual B cells into germinal centers, cell division and death. The total rate of events $\lambda$ is given by

$$\lambda = \lambda_{\text{immigration}} + \lambda_{\text{division}} + \lambda_{\text{death}} \tag{1}$$

where the terms on the right-hand side correspond to the rate of each kind of event (mutation is associated with cell division and is therefore not an independent event). The algorithm consists of drawing the time to the next event by sampling from an exponential distribution with rate $\lambda$. Once an event has occurred, we a make a second draw to determine its type. The probability for each type of event in this second draw is proportional to the corresponding event-specific rate (e.g., the probability that the next event is a cell division is $\lambda_{\text{division}}/\lambda$). After determining the event type, we update event rates and draw the time to the next event, and so on, until a maximum time $t_{\text{max}}$ is reached. For each germinal center, we record the number of cells in each B cell lineage and the V alleles used by the lineages at the end of day 1 and then every 5 days starting on day 5.

Immigration of B cells into germinal centers is restricted to an initial period with duration $t_{\text{imm}}$. Parameter $I_{\text{total}}$ controls the expected number of lineages that enter each germinal center during that time (each recruited B cell is the founder of an individual lineage). Given those parameters, we let $\lambda_{\text{immigration}}$ be a linearly decreasing function over time reaching 0 at $t_{\text{imm}}$, with intercept and slope chosen such that $I_{\text{total}}$ lineages are expected to enter each GC by that point ($\lambda_{\text{immigration}}$ then remains at 0 until the end of the simulation).

When immigration occurs, we randomly sample a single immigrant from a newly generated recruitment pool of 1,000 naive cells whose V alleles are drawn with replacement from the naive repertoire. For each member of the recruitment pool, we sample an affinity value based on the naive affinity distribution associated with its V allele. The probability that each cell in the recruitment pool is chosen as the new immigrant is then proportional to its affinity. By default, all alleles have the same normal affinity distribution with mean $a$ and standard

deviation $\sigma$ (we sample from the associated truncated distribution to avoid negative values). Depending on the scenario, naive B cells using specific V alleles may have a different distribution with mean $a + s\sigma$ and the same standard deviation $\sigma$ (i.e., the increment in the mean is expressed in units of the baseline standard deviation).

The rate of cell divisions depends on the total number of cells inside the germinal center, $N$, and on the rate of cell division for each individual cell, $\mu(N)$:

$$\lambda_{\text{division}}(N) = N \times \mu(N) \tag{2}$$

To represent competition for antigen, $\mu(N)$ decreases with $N$ so that it equals a fixed per-cell death rate $\delta$ when the population is at carrying capacity ($N = K$):

$$\mu(N) = \mu_{\text{max}} \times \exp\left[\frac{N}{K}(\ln \delta - \ln \mu_{\text{max}})\right] \tag{3}$$

Once a division event occurs, we randomly sample a B cell to divide. The probability that a B cell is chosen is proportional to its affinity. Each dividing B cell has some probability of having a mutation that changes affinity by a normally distributed amount with mean 0 and standard deviation given by $\beta\sigma$ (i.e., in units of the baseline naive affinity standard deviation $\sigma$) (affinity is set to 0 if the mutation produces a negative value). By default, all B cells have the same baseline mutation rate regardless of their germline V allele. Depending on the scenario, naive B cells using specific V alleles may have this rate multiplied by a factor $\gamma$.

Finally, with a fixed per-cell death rate, the population-level death rate is given simply by

$$\lambda_{\text{death}}(N) = N\delta \tag{4}$$

## Sensitivity of model results to truncation of naive affinity distributions

For the default values used in the analyses ($a = \sigma = 1$), the choice of a truncated normal distribution to model naive affinity distributions causes the effective variance to increase as the mean increases (since fewer values run into the truncation). To test if our results were sensitive to this effect, we repeated the simulations of the high-affinity scenario after setting $\sigma = 0.1$ (thus reducing the fraction of truncated values in the baseline distribution from 0.16 to effectively zero and causing the variance to remain approximately constant as $s$ increases). Because the increase in mean affinity ($s$) and the standard deviation of mutational effects ($\beta$) are expressed as multiples of $\sigma$, we re-scaled those parameters to explore the same absolute values as in the main analysis (since $\sigma$ was divided by 10, we multiplied $s$ and $\beta$ by 10).

We found the same behavior as in the main analysis, with germline-encoded specificities leading to persistent similarity if $s$ was sufficiently large relative to $\beta$ and the mutation rate, and decreasing similarity if $s$ was relatively small (S23 Fig). Absolute increases of 0.5 and 1 in mean affinity for high-affinity alleles, which result in very low or zero correlation between individuals in the original analysis (Fig 3), now result in strong correlations. This difference is due to the fact the higher value of $\sigma$ in the original simulations leads to stronger overlap between the affinity distributions of high- and low-affinity alleles.

## Experimental infection of mice with an influenza A/H1N1 virus

We infected 40 8-week-old female C57BL/6 mice (Jackson Laboratories) weighing 20–22g (8 for each time point) intranasally with 0.5 LD$_{50}$ of a mouse-adapted pandemic H1N1 strain (A/Netherlands/602/2009) in a total of 30 $\mu$L of PBS under full anesthesia. In addition, two controls for each time point were given PBS only.

## Tissue processing, cell sorting and nucleic acid extraction

We prepared single cell suspensions from the mediastinal lymph node, spleen and both femurs harvested at the indicated time points. B cells were first enriched from the splenocyte suspension by MACS (magnetic activated cell sorting) using the Pan B cell Isolation Kit (Miltenyi Biotec), followed by staining for FACS (fluorescence activated cell sorting). The lymph node and bone marrow cells were directly stained for FACS. Antibodies used for sorting were anti-B220 (clone RA3-6B2, Biolegend), IgD (clone 11-26c.2a, Biolegend), anti-CD4 (clone RM4-5, Biolegend), anti-CD8 (clone 53-6.7, Biolegend), anti-CD38 (clone 90, Biolegend), anti-CD95 (clone Jo-2; BD Biosciences), anti-CD138 (clone 281-2, Biolegend), anti-F4/80 (clone BM8, Biolegend), anti-GL7 (clone GL7, BD Biosciences), anti-Sca-1 (clone D7, Biolegend), and anti-TER-119 (clone TER-119, Biolegend). Antibody stainings were preceded by adding Fc block (anti-CD16/CD32; clone 2.4G2, BD Biosciences). For sorting, the cells were first gated on size and granularity (forward and side scatter, respectively) to exclude debris, followed by doublet exclusion. We sorted naive (IgD+B220+), plasma (IgD-Sca-1hiCD138hi), memory (IgD-B220+CD95-CD38hi) and germinal center (IgD-B220+CD95+ CD38loGL-7+) cells after excluding cells expressing CD4, CD8, TER-199 or F4/80 (to exclude T cells, erythroid cells and macrophages). After spinning down cells and removing the PBS supernatant, we extracted DNA and RNA from the cell pellets using the AllPrep DNA/RNA Mini Kit (Qiagen), according to the manufacturer's protocol. All samples were kept frozen until sequenced.

## B cell receptor sequencing

We generated immunoglobulin heavy chain (IGH) DNA libraries from complementary DNA generated from 10–500 ng of total RNA using Superscript III (Invitrogen) reverse transcriptase and random hexamer primers. For PCR amplifications, we used multiplexed primers targeting the mouse framework region 1 (FR1) of IGHV in combination with isotype-specific primers targeting constant region exon 1 of IgA, IgD, IgE, IgG, or IgM (Tables 2 and 3). We performed separate PCR reactions for each isotype to avoid formation of inter-isotype chimeric products. We barcoded each sample with 8-mer primer-encoded sequences on both ends of the amplicons and performed PCR amplification in two steps. First, we generated amplicons using primers with the partial Illumina adapter, the sample-specific barcode and the locus-specific sequence. In the second step, we performed another PCR to complete the Illumina adapter sequence and to ensure final products were not amplified to saturation. We purified pooled products by agarose gel electrophoresis and extraction. We used a 600 cycle v3 kit to sequence products using an Illumina MiSeq instrument.

We estimated the rate at which errors were introduced during amplification and sequencing by comparing the sequenced reads with the reference sequence for the corresponding isotype. Because the constant region does not undergo somatic hypermutation, we counted each mismatch between the end of the J gene and the beginning of the conserved region primer as an error introduced by sequencing and amplification. Based on 187,500 errors found out of 104,092,368 bases analyzed, we estimated the error rate to be 1.80 mutations per thousand bases (95% binomial CI 1.79–1.81).

## ELISA

We coated 96-well ELISA plates (Thermo Fisher Scientific) overnight at 4°C with eight hemagglutination units (HAU) of virus in carbonate buffer. We used horseradish peroxidase (HRP)-conjugated goat anti-mouse IgG antibody (Southern Biotech) to detect binding of serum antibodies, followed by development with Super Aquablue ELISA substrate (eBiosciences). We measured absorbance at 405 nm on a microplate spectrophotometer (Bio-Rad). We analyzed

**Table 2. Primers for mouse heavy chain B cell receptors.**

| Primer name | Sequence |
|---|---|
| P7-VH1-MsFR1-A | CCTGGGGCTTCAGTGA |
| P7-VH1-MsFR1-B | GCCTGGGACTTCAGTGA |
| P7-VH1-MsFR1-C | CCTGGGGCCTCAGTGA |
| P7-VH1-MsFR1-D | GCCTGGGGCTTCAGTAA |
| P7-VH2-MsFR1 | CCCTCACAGAGCCTGT |
| P7-VH3-MsFR1 | CTTCAGGAGTCAGGACCT |
| P7-VH5-MsFR1-A | GTCCCTGAAACTCTCCTGTG |
| P7-VH5-MsFR1-B | GCCTGGAAGGTCCGT |
| P7-VH5-MsFR1-C | GTCCCTGAAACTCTCCTG |
| P7-VH7-MsFR1 | TTCTCTGAGACTCTCCTGTG |
| P7-VH9-MsFR1 | TGGAGAGACAGTCAAGATCTCC |
| P7-VH10-MsFR1 | GATTGGTGCAGCCTAAAGG |
| P7-VH11-MsFR1 | GCTTGGTGCAACCTGG |
| P7-VH12-MsFR1 | TGCTGTCATCAAGCCATCA |
| P7-VH14-MsFR1 | AGTCAAGTTGTCCTGCA |
| Ms-Tim-IgM | GGGAAGACATTTGGGAAGGAC |
| Ms-Tim-IgD | TGAGAGGAGGAACATGTCAG |
| Ms-Inner-IgG1 | GCTCAGGGAAATAGCCCTTGAC |
| Ms-Inner-IgG2 | GCTCAGGGAAATAACCCTTGAC |
| Ms-Inner-IgG2b | ACTCAGGGAAGTAGCCCTTGAC |
| Ms-Inner-IgG3 | GCTCAGGGAAGTAGCCTTTGAC |
| Ms-Tim-IgA | GTCAGTGGGTAGATGGTGG |
| Ms-Tim-IgEc | CCAGGCAGCCCAGGGTCATGG |

**Table 3. PCR conditions.**

| 1st PCR | | 2nd PCR | |
|---|---|---|---|
| usual initial mix | | | |
| | per rxn | MM | 10 |
| 10x buffer | 3 $\mu$L | Q mix | 4 |
| MgCl2 | 1.8 | f primer | 0.4 |
| 2mM dNTP | 3 | r primer | 0.4 |
| v primer | 3 | template | 0.5 |
| c primer | 3 | H2O | 4.7 |
| template | 4$\mu$L (200ng total) | | |
| Taq | 0.3 | **2nd PCR cycle** | |
| H2O | 11.9 | usual illumina cycling | |
| total | 30 | | |
| | | 95˚C | 15 min |
| **1st PCR cycle** | | 95˚C | 30s (× 12 cycles) |
| | | 60˚C | 45s |
| 94˚C | 7 min | 72˚C | 1.5 min |
| 94˚C | 30s (× 35 cycles) | 72˚C | 10 min |
| 56˚C | 45s | | |
| 72˚C | 1.5 min | | |
| 72˚C | 10 min | | |

serum samples starting at a top dilution of 1:20 (PBS controls and day 8 animals) or 1:1000 (all other samples), followed by 2-fold dilutions in 8 (PBS controls and day 8 animals) or 16 steps. We determined the end titer as the last dilution point with an OD value of $> 2x$ the blank average OD value for each respective plate.

## Estimating the frequencies of V alleles and B cell lineages

We used *partis* v0.16.0 [78–80] to partition sequences into lineages and identify the germline alleles used by each lineage's naive ancestor. Briefly, *partis* identifies lineages by comparing the probability that two clusters of sequences came from a single rearrangement event with the probability that they came from separate rearrangement events. Sequences in the same lineage must have the same germline V and J segments, but *partis* does not require a minimum degree of similarity in CDR3 for sequences to be part of the same lineage. For the main analysis, we used the reference set of C57BL/6 alleles curated by the Open Germline Receptor Database (available as an option in *partis*) based on sequences from [81], disabling the inference of novel alleles that *partis* performs by default. We also repeated the analyses of germline allele frequencies using *partis* with the IMGT mouse reference set (with and without inference of novel alleles) and again using IgBLAST v.1.3.0. [82]. We used the fraction of reads corresponding to each allele as a proxy for the frequency of that allele in each B cell population. To reduce the error in frequency estimates, we excluded B cell populations with fewer than 100 reads in a mouse. Since we did not barcode individual cells or RNA molecules during sequencing, the number of reads with a particular sequence reflects not only the number of B cells with that sequence but also their transcription levels. We also repeated the analyses using the number of unique sequences to estimate the abundance of each lineage or allele (i.e., counting multiple identical reads from the same mouse, tissue, cell type and isotype only once).

We measured the size of each lineage in each lymph-node B cell population as the number of reads from that lineage in that population (as opposed to the number of reads in the lineage across all cell types and tissues). B cell lineages were mostly confined to a single tissue and usually dominated by a single cell type (S22 Fig; note that the partitioning of sequences into lineages was agnostic to cell type and tissue).

While we initially considered Dump-IgD+B220+ cells as naive cells, we noticed that many sequences obtained from them were extensively mutated relative to their inferred germline genes and were also inferred to be part of large clonal expansions. To exclude reads originating from non-naive B cells sorted as IgD+B220+, we considered a read as likely coming from a naive cell if it met all of the following criteria: 1) it came from IgD+B220+ samples; 2) its isotype was IgM or IgD; 3) it belonged to a clone that had a single unique sequence across its reads (and the reads all came from IgD+B220+ samples), and 4) that sequence had at most two nucleotide mutations in the V gene region. To compute naive frequencies, we pooled sequences meeting those criteria across all tissues. When computing experienced-to-naive frequency ratios, we adjusted the frequencies of germline alleles that were sampled in an experienced B cell population but not in naive B cells, since those alleles must have been present in naive B cells even though they were not sampled. In those cases, we imputed a single sequence to the allele in the naive repertoire then recalculated naive allele frequencies accordingly. When computing frequency deviations from the naive repertoire, we excluded mice with fewer than 100 naive reads even if the corresponding experienced population had more than 100 reads.

To test if our results were robust to uncertainty in the identification of naive B cells in our data, we alternatively estimated V allele frequencies from naive B cells (CD138-CD19+IgD+ +IgM+CD23++ CD21+PI-) sampled by [52] from the spleen of healthy C57BL/6 mice. For

these data, we processed raw paired-end reads using *presto* v.0.6.2. [83], then used *partis* to identify germline V alleles for a random sample of 20,000 sequences per mouse. V allele frequencies (measured by the fraction of total reads assigned to each gene in each mouse) were positively correlated between this independent dataset and the designated naive populations from our data (mean Pearson correlation coefficient between pairs of mice from each dataset = 0.61, interquartile range 0.52–0.76). We repeated the analysis of pairwise frequency-deviation correlations over time after replacing naive frequencies in our mice with the average frequency of each gene in the [52] dataset, preserving the number of reads in each mouse. When calculating the average allele frequencies in the alternative data set, we artificially assigned a single read to alleles present in our mice but absent from the alternative data set (since genes present in the experienced cells cannot be entirely missing from the naive repertoire) and re-normalized frequencies so they would sum to 1.

### Identifying overrepresented germline alleles

To determine which germline alleles were consistently overrepresented in experienced B cell populations relative to the naive repertoire, we compared the frequency deviations for each germline allele (separately for each type of B cell) with the distribution expected if alleles were sampled based on naive frequencies alone (maintaining the observed the number of sequences in each mouse). For each germline allele, we then counted the number of mice with stronger deviations from the naive repertoire than expected under this null distribution (using a 95% bootstrap confidence interval).

### Mutability analysis of germline alleles

To estimate the mutability of mouse germline V alleles, we used the RS5NF mutability scores estimated by [54] using non-functional mouse kappa light-chain sequences and implemented in R package *shazam*. These scores describe the relative mutability of all possible 5-nucleotide motifs. We estimated the mutability of each framework region (FR) and complementarity-determining region (CDR) as the average score across motifs in the region. We then calculated an average for all FRs weighted by the length of each FR, and similarly for CDRs. We used *igblast* v1.14.0 to identify the FRs and CDRs of each germline V allele sequence identified by *partis*.

### Measuring CDR3 sequence similarity

We compared the amino acid sequence similarity and biochemical similarity of pairs of CDR3 sequences sampled from different mice and matched either for length alone or both for length and V allele. To limit the number of comparisons, we proceeded as follows. For each pair of mice, we chose one mouse and sampled 500 sequences of the same cell type. For each sequence length represented in this sample, we paired sequences from the first sample with randomly chosen sequences of the same length from the second mouse. If matching sequences both for length and V allele, we did this second sampling separately for each combination of V allele and sequence length present in the first sample. This procedure matches sequences while preserving the length distribution (or the joint distribution of length and V alleles) in the first sample.

We measured amino acid sequence similarity as the proportion of sites with the same amino acid in both sequences. Following previous work [84, 85], we measured biochemical similarity as the proportion of sites in which the amino acids of both sequences belonged to the same category in the classification by [86]: hydrophobic (F, L, I, M, V, C, W), hydrophilic (Q, R, N, K, D, E) or neutral (S, P, T, A, Y, H, G).

## Supporting information

**S1 Fig. Simulated correlation in germline allele frequencies and experienced-to-naive ratios between different numbers of simulated individuals.** We measured germline allele frequencies across germinal centers in each individual and computed the ratio between these frequencies and the allele frequencies in the naive repertoire. Points and vertical bars represent the median and the 1st and 4th quartiles of these correlations across all pairs (i.e., the bars represent true variation in simulated outcomes and not the uncertainty in the estimate of the median). We let 5 germline alleles have the mean of their naive affinity distribution increased by $s = 1.5$ relative to the baseline. We set the mutation rate to 0.01 mutations per B cell division and $\beta = 4$. Other parameters were set to the default values in Table 1.
(TIF)

**S2 Fig. Correlation in germline allele frequencies in the scenario where the same 5 germline alleles have higher expected affinity than the others in all individuals.** For each parameter combination, we simulated 100 individuals with varying numbers of germinal centers. We measured germline allele frequencies across germinal centers in each individual and computed the correlation in those frequencies between all pairs of individuals. Points and vertical bars represent the median and the 1st and 4th quartiles of these correlations across all pairs (i.e., the bars represent true variation in simulated outcomes and not the uncertainty in the estimate of the median). For these simulations, we assumed 15 germinal centers per individual. Other parameter values are as in Table 1.
(TIF)

**S3 Fig. Combined frequency of high-affinity alleles within germinal centers in simulations.** Columns show the distribution across germinal centers early (10 days) and late (50 days) in the response. Rows show different somatic hypermutation rates (mutations per B cell per division). For each row, we simulated 100 individuals, each with 30 germinal centers. The same five alleles in all individuals were chosen to have their average naive affinity increased by $s = 2$. We set $\beta = 4$ and other parameters to the values in Table 1.
(TIF)

**S4 Fig. Simulations in the scenario where some alleles have higher expected naive affinity than others.** For each parameter combination, we simulated 100 individuals with varying numbers of germinal centers. We measured germline allele frequencies across germinal centers in each individual and computed the ratio between these frequencies and the allele frequencies in the naive repertoire. We then computed the correlation in these experienced-to-naive-ratios between all pairs of individuals. Points and vertical bars represent the median and the 1st and 4th quartiles of these correlations across all pairs (i.e., the bars represent true variation in simulated outcomes and not the uncertainty in the estimate of the median). For these simulations, we assumed 15 germinal centers per individual. Other parameter values are as in Table 1.
(TIF)

**S5 Fig. Combined frequency of high-mutability alleles within germinal centers in simulations.** Columns show the distribution across germinal centers early (10 days) and late (50 days) in the response. Rows show different baseline mutation rates (affinity-changing mutations per B cell per division). For each row, we simulated 100 individuals, each with 30 germinal centers. The same five alleles in all individuals were chosen to have their baseline mutation rate increased by $\gamma = 6$. We set $\beta = 4$ and other parameters to the values in Table 1.
(TIF)

**S6 Fig. Simulations in the scenario where the same 5 germline alleles have a higher mutation rate than the others in all individuals.** For each parameter combination, we simulated 100 individuals with varying numbers of germinal centers. We measured germline allele frequencies across germinal centers in each individual and computed the ratio between these frequencies and the allele frequencies in the naive repertoire. We then computed the correlation in these experienced-to-naive-ratios between all pairs of individuals. Points and vertical bars represent the median and the 1st and 4th quartiles of these correlations across all pairs (i.e., the bars represent true variation in simulated outcomes and not the uncertainty in the estimate of the median). For these simulations, we assumed 15 germinal centers per individual. $\gamma$ represents the factor by which the baseline mutation rate is multiplied in high-mutation alleles, while $\beta$ represents the standard deviation of mutation effect size on affinity. Other parameter values are as in Table 1.
(TIF)

**S7 Fig. Representative flow cytometry plots for gating strategy for cell sorting.** Representative plots for gating strategy for mediastinal lymph nodes (A), spleen (B), and bone marrow (C). Debris and doublets were first gated out (not shown), followed by exclusion of cells expressing CD4, CD8, TER-119, and/or F4/80 (DUMP). Lymph node (A) and spleen (B) IgD +DUMP- cells that were positive for B220 were sorted as naïve B cells. IgD-DUMP- cells that were Sca-1hiCD138hi were sorted as plasma cells. Sca-1lo/-CD138lo/- cells that were also B220+ were further gated and sorted as memory cells (CD95-CD38hi) or germinal center B cells (CD95+CD38lowGL7+). From bone marrow, we only sorted naïve B cells (IgD+DUMP-B220+) and plasma cells (IgD-DUMP- Sca-1hiCD138hi) (C). These representative plots show tissue from one mouse 56 days after primary infection.
(TIF)

**S8 Fig. Number of cells sorted from different tissues in mice infected with influenza and in uninfected controls.** Infected mice were subject to one or two infections and sacrificed at 8, 16, 24, 40 or 56 days after primary infection. Mice from the last two time points were given a second infection 32 days after the first one.
(TIF)

**S9 Fig. Number of germline V alleles detected in each mouse under different annotation tools and reference allele sets.** Each circle represents the observed value for an individual mouse, with the associated rarefaction curve indicating the expected number of alleles if only a random subset of the mouse's sequences had been sampled.
(TIF)

**S10 Fig. Evidence of increased B cell evolution and competition over time in the infected mice.** (**A**) Serum antibody titers against the infecting strain measured by ELISA. (**B**) Total fraction of reads in influenza-induced populations represented by the ten largest B lineages in each mouse. The ten largest lineages were chosen based on the number of reads each lineage had in the respective cell type in the lymph node (not the total number of reads each lineage had across all tissue and cell types). (**C**) Fraction of lineages with at least one amino acid mutation at frequency 50% of higher in the lineage (top panel), and the average number of such high-frequency mutations per lineage within each mouse (bottom panel). Mutation frequencies in each lineage were calculated relative to the lineage's number of reads in the respective tissue and cell type combinations. For these calculations, only lineages with at least 10 reads were considered.
(TIF)

**S11 Fig. Increasing dominance by mutated clones over time is evident in lymph nodes but not in other tissues.** Fraction of clones with at least one amino acid mutation at frequency 50% of higher (top panel) and the average number of such high-frequency mutations per clone (bottom panel) for different cell types and tissues. Mutation frequencies in each clone were calculated relative to the clone's number of reads in the respective tissue and cell type combinations (not the total number of reads in the clone across all subtypes and tissues). For each combination of cell type and tissue, each point corresponds to a mouse. Only clones with at least ten reads were considered.
(TIF)

**S12 Fig. Frequency of the 10 most common V alleles in lymph node plasma cells of infected mice.** Each panel represents an individual mouse. The arrows go from each allele's frequency in the naive repertoire to its frequency in lymph node plasma cells. Mouse 40-7, which was sacrificed 8 days after the secondary infection, was considered a day-8 primary-infection mouse because it showed no signs of infection after the first inoculation and had ELISA titers similar to those of day-8 infected mice.
(TIF)

**S13 Fig. Frequency of the 10 most common V alleles in lymph node germinal center cells of infected mice.** Each panel represents an individual mouse. The arrows go from each allele's frequency in the naive repertoire to its frequency in lymph node plasma cells. Mouse 40-7, which was sacrificed 8 days after the secondary infection, was considered a day-8 primary-infection mouse because it showed no signs of infection after the first inoculation and had ELISA titers similar to those of day-8 infected mice.
(TIF)

**S14 Fig. Frequency of the 10 most common V alleles in lymph node memory cells cells of infected mice.** Each panel represents an individual mouse. The arrows go from each allele's frequency in the naive repertoire to its frequency in lymph node plasma cells. Mouse 40-7, which was sacrificed 8 days after the secondary infection, was considered a day-8 primary-infection mouse because it showed no signs of infection after the first inoculation and had ELISA titers similar to those of day-8 infected mice.
(TIF)

**S15 Fig. Frequency deviations from the naive repertoire for germline heavy-chain V alleles IGHV14-4\*01, IGHV1-82\*01 and IGHV1-69\*01 at different time points and in different cell types.** We measured frequency deviations as the ratio of the experienced-to-naive frequencies in each population. Each point represents a mouse with at least 100 sequences sampled from the corresponding experienced population and from the naive repertoire. Deviations from the naive repertoire are colored based on whether they are different from a null distribution obtained by bootstrapping experienced frequencies from the naive repertoire ($n = 500$ replicates) based on a 95% confidence interval test.
(TIF)

**S16 Fig. Correlations between mice in germline V allele frequencies and their deviations from the naive repertoire in sensitivity analyses show robustness of results to germline annotation tool and allele reference set.** Each colored point represents a pair of mice with at least 100 reads each in the respective B cell population. The horizontal bars indicate the observed median across mouse pairs, whereas the black circles and black vertical bars indicate the bootstrap average and 95% confidence interval for the median in a null model with V alleles randomly assigned to each B cell lineage ($n = 500$ randomizations). We computed

correlations using Pearson's coefficient and measured frequency deviations as the ratio between a V allele's frequency in an influenza-induced population and its frequency in the naive repertoire.
(TIF)

**S17 Fig. Distribution of predicted mutability across V alleles (A), and correlations between predicted mutability and their frequency deviations from the naive repertoire (B).** For each framework region (FR) and complementarity-determining region (CDR), we computed the average RS5NF mutability score from [54] across all 5-nucleotide motifs. In **B**, we computed an average across FRs weighed by the length of each FR, and similarly for CDRs. Each circle represents a mouse with at least 100 sequences each in the naive and experienced populations. Correlations were measured using Pearson's coefficient.
(TIF)

**S18 Fig. Number of high frequency mutations as a function of clone rank in lymph node germinal center cells (top) and lymph node plasma cells (bottom). Each point represents a clone**. Mice from each time point (8, 16, 24, 40 and 56 days after primary infection with influenza) were pooled together in each panel. Clone rank was determined based on the number of reads each clone had in the respective population (lymph node germinal center cells or lymph node plasma cells), not the total number of reads in the clone across all cell types and tissues (the largest clone was assigned rank 1). The solid line is a locally estimated scatterplot smoothing (LOESS) spline.
(TIF)

**S19 Fig. Probability that two B cell lineages sharing the same heavy chain V allele have high-frequency mutations in common.** Panels represent B cell types from the lymph node of mice infected with influenza virus (GC: germinal center cells, PC: plasma cells, mem: memory cells). High-frequency mutations were those with a frequency of 50% within the lineage (considering lineage reads in each cell type). The numbers above the bars indicate the number of lineage pairs being compared (pairs were from either the same mouse or difference mice). We restricted the analysis to lineages with at least 10 reads.
(TIF)

**S20 Fig. Similarity of CDR3 sequence pairs sampled from different mice and matched for the same length (top) or the same length and the same V allele (bottom).** Boxplots show the distribution across sequence pairs from all mouse pairs for each time point (separately for different cell types). Values that fall outside 1.5 times the inter-quartile range are shown as individual points.
(TIF)

**S21 Fig. Diversity of CDR3 sequences in the naive repertoire for each germline heavy chain V allele.** For each mouse, we identified all pairs of sequences in the naive repertoire that had the same germline allele and the same CDR3 length. For each pair, we computed the fraction of sites that had different amino acids (sequence dissimilarity, bottom), or the fraction with amino acids in different biochemical classes (top). The boxplots show the distribution of values for each V allele pooled across mice. Germline alleles consistently overrepresented in day-8 lymph node plasma cells of mouse infected with influenza are highlighted in red.
(TIF)

**S22 Fig. Fraction of clones (with at least 10 reads) that have 90% or more reads from a single tissue (left) or 90% or more reads from a single cell type (right).** Each point represents a

mouse.
(TIF)

**S23 Fig. Simulations of the high-affinity scenario with a lower baseline standard deviation for the naive affinity distribution ($\sigma = 0.1$).** For each parameter combination, we simulated 100 individuals with varying numbers of germinal centers. We measured germline allele frequencies across germinal centers in each individual and computed the ratio between these frequencies and the allele frequencies in the naive repertoire. We then computed the correlation in these experienced-to-naive-ratios between all pairs of individuals. Points and vertical bars represent the median and the 1st and 4th quartiles of these correlations across all pairs (i.e., the bars represent true variation in simulated outcomes and not the uncertainty in the estimate of the median). For these simulations, we assumed 15 germinal centers per individual. Other parameter values are as in Table 1.
(TIF)

## Acknowledgments

This work was completed in part with resources provided by the University of Chicago Research Computing Center. Lauren McGough, Christopher Russo and four anonymous reviewers made helpful comments and suggestions to the manuscript.

## Author Contributions

**Conceptualization:** Marcos C. Vieira, Patrick C. Wilson, Sarah Cobey.

**Data curation:** Marcos C. Vieira.

**Formal analysis:** Marcos C. Vieira.

**Funding acquisition:** Scott D. Boyd, Patrick C. Wilson, Sarah Cobey.

**Investigation:** Marcos C. Vieira, Anna-Karin E. Palm, Christopher T. Stamper, Micah E. Tepora, Khoa D. Nguyen, Tho D. Pham.

**Methodology:** Marcos C. Vieira, Anna-Karin E. Palm, Patrick C. Wilson, Sarah Cobey.

**Project administration:** Scott D. Boyd, Patrick C. Wilson, Sarah Cobey.

**Resources:** Scott D. Boyd, Patrick C. Wilson, Sarah Cobey.

**Software:** Marcos C. Vieira.

**Supervision:** Scott D. Boyd, Patrick C. Wilson, Sarah Cobey.

**Validation:** Marcos C. Vieira.

**Visualization:** Marcos C. Vieira, Anna-Karin E. Palm.

**Writing – original draft:** Marcos C. Vieira, Sarah Cobey.

**Writing – review & editing:** Marcos C. Vieira, Anna-Karin E. Palm, Christopher T. Stamper, Micah E. Tepora, Khoa D. Nguyen, Tho D. Pham, Scott D. Boyd, Patrick C. Wilson, Sarah Cobey.

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
