## [Decision Letter · Decision Letter 0]

24 Mar 2023

Dear Dr. Costa Vieira,

Thank you very much for submitting your manuscript "Increasingly divergent responses to infection in mice suggest B cell evolution is not constrained by germline-encoded specificities" for consideration at PLOS Pathogens. As with all papers reviewed by the journal, your manuscript was reviewed by members of the editorial board and by several independent reviewers. In light of the reviews (below this email), we would like to invite the resubmission of a significantly-revised version that takes into account the reviewers' comments.

The report presents computational modeling and wet-lab evidence that germline encoded antibody sequences are an early determinant of the B cell response to influenza virus infection but that over time the response diversifies. The reviewers commented that the report was well-written with potentially interesting and exciting findings that may be of broad interest. Nevertheless, there was a mixed level of enthusiasm and a number of concerns were identified by each reviewer. Most notably, these include the appropriate use of statistical analyses (R1/R2,), the need to include gating strategies (R1/2), consistencies with public responses to influenza virus in humans (R1/4), accounting for the rates of SHM (R2/R3/R4), sample sizes (20) used for the modeling (R3), the validity of the conclusions and accuracy of the title (R3), and the binding specificities of the antibodies (R4). Although there are an array of issues it is likely that all of the main concerns are quite addressable and that taking these points into consideration will improve the impact and quality of the study.

We cannot make any decision about publication until we have seen the revised manuscript and your response to the reviewers' comments. Your revised manuscript is also likely to be sent to reviewers for further evaluation.

Sincerely,

Allan J Zajac

Academic Editor

PLOS Pathogens

Ronald Swanstrom

Section Editor

PLOS Pathogens

Kasturi Haldar

Editor-in-Chief

PLOS Pathogens

orcid.org/0000-0001-5065-158X

Michael Malim

Editor-in-Chief

PLOS Pathogens

orcid.org/0000-0002-7699-2064

The report presents computational modeling and wet-lab evidence that germline encoded antibody sequences are an early determinant of the B cell response to influenza virus infection but that over time the response diversifies. The reviewers commented that the report was well-written with potentially interesting and exciting findings that may be of broad interest. Nevertheless, there was a mixed level of enthusiasm and a number of concerns were identified by each reviewer. Most notably, these include the appropriate use of statistical analyses (R1/R2,), the need to include gating strategies (R1/2), consistencies with public responses to influenza virus in humans (R1/4), accounting for the rates of SHM (R2/R3/R4), sample sizes (20) used for the modeling (R3), the validity of the conclusions and accuracy of the title (R3), and the binding specificities of the antibodies (R4). Although there are an array of issues it is likely that all of the main concerns are quite addressable and that taking these points into consideration will improve the impact and quality of the study.

Reviewer's Responses to Questions

**Part I - Summary**

Reviewer #1: Summary:

The authors measure the degree to which non-random, antibody V gene encoded responses progress overtime after infecting mice with influenza virus. Germline encoded or public antibody responses to antigen are seen in humans and can sometimes offer genetically hardwired solutions to engage conserved sites of vulnerability on hypervariable viruses including influenza. The authors apply both wetlab experiments and computational models to demonstrate that germline V gene usage dominates the early timepoints of B cell expansion in germinal centers and plasma cell responses but becomes increasingly supplanted by non-public lineages. The authors surmise that germline encoded specificities offer rapid recognition of specific antigens, but that this deterministic predictability of outcome is lower over more sustained humoral immune reactions.

Reviewer #2: The article “Increasingly divergent responses to infection in mice suggest B cell evolution is not constrained by germline-encoded specificities” by M. Viera et al. is a study investigating the constraints of Ig Variable gene usage within B cell immune responses. The authors uses modeling of B cell behavior in immune responses as a function of V gene usage to evaluate the potential roles of ‘hard coded’ reactivity influenced by CDRs 1 and 2 which are germline encoded, mutability of V genes based on bioinformatic analysis of V gene sequences, and germinal center dynamics to evaluate the potential contribution of these factors to productive B cell responses. To evaluate the relative contributions of stochastic events during B cell antigen receptor diversification, the authors compare the results of these models with B cell antigen receptor sequences from the draining lymph node, spleen, and bone marrow of mice harvested longitudinally following infection with murine adapted Influenza virus.

Based on the results of this study, the authors conclude that even in situations where specific V genes are predominantly expanded early in B cell response, due to an increased frequency of antigen reactivity within certain V gene expressing receptors, continued somatic mutation and affinity maturation events result in the diversification of the antigen reactive repertoire skewing away from any Ig V gene bias.

This is a well written manuscript although there are concerns:

Reviewer #3: This is a very interesting and exciting paper that brings new evidence to bear on the question of how important germline variation is likely to be on the adaptive potential of the humoral response. The introduction and discussion are well-written. I particularly liked the pairing of simulation and experimental approaches and thought it they did indeed complement each other.. However, I am not convinced that the evidence here strongly supports the central claims of the paper. Below I discuss some critiques; for clarity, I have divided these into comments on the model and comments on the experiment.

Reviewer #4: In this manuscript, Viera and colleagues set out to study how the germline gene repertoire might constrain downstream antibody responses. They study this very important question both using simulations and experimental mouse data.

Strength: The study is extensive and I appreciate that both simulation as well as experimental data have been leveraged to investigate the research question.

Weakness: I have concerns about the simulation model used as well as the experimental data generated.

**Part II – Major Issues: Key Experiments Required for Acceptance**

Reviewer #1: Overall:

This is an interesting study that attempts to wholistically and experimentally define the germline antibody contribution to antibody response as it progresses over time. The key finding that initial germline biasing is eventually supplanted by non-public antibody responses represents an important ‘dilution’ principle that will be of wide interest to the field. However, critical statistically comparisons are missing and need to be included to solidify this conclusion. Moreover, the authors need to address why public antibody responses in humans appear to ‘escape’ this dilution effect. Essentially all of them have been identified by heavy biasing within B cell stages observed at later stages of development, post affinity maturation.

Major:

1) Figure 3C and Figure 4A, are key results indicating that V gene usage dominates early, but then becomes supplanted by non-public B cell lineages, in some cases as the primary response progresses, and also, in some cases post-secondary exposure to the virus. However, there is no statistical analyses performed to define/verify these differences and conclusions.

2) Line 259. Author write “These results suggest that while germline-encoded advantages may strongly shape the early B cell response, they do not predict B cell fitness in the long run.”. This seems to be at odds with public antibody responses seen humans. Just about all of them have been discovered by the identification of over-represented V gene usage at post-affinity maturation B cell stages. How do the authors account for this if their model is to be generalized? A well described case of influenza responses includes IGHV1-69 usage, where it dominates both germline reactivity to antigen and is over represented in later stages of B cell expansion. Could it be such that different germline encoded responses dilute to different degrees?

3) Figure 3C. The authors should see whether this relationship holds if the pairing comparison is performed between the different mice (expanded in mouse 1 vs expanded in mouse 2-6 etc). This will test how the truly public (donor-independent) V gene usage changes over time.

Reviewer #2: 1. The authors appear to conflate individual V gene segments and alleles throughout the manuscript. C57Bl6 mice, being an inbred strain should only express a single allele of each IGHV gene.

2. The authors should include B cell subset sort plots from tissues in supplemental figures.

3. The diversity of CDR3 sequences across V genes should be assessed. – the observation that some V genes are more rapidly recruited into influenza responses may also be explained by differences in diversity across V genes in the pre-existing repertoire.

4. B cell lineage should be clearly defined (commonly defined by V and J gene identity, CDR3 length, and a percentage nucleic acid sequence similarity)

5. Do the rates of somatic mutation in experimental date derived from repertoire data reflect mutability scores?

Reviewer #3: The model

My core critique of this model is that the stated goal (line 157) is not to make quantitative predictions based on realistic parameter values. However, the main conclusions one would draw from the model seem to depend critically on the actual values. For instance, it is clear from the simulations that if rates of SHM are low enough, then germline-based affinity matters a lot whereas if rates are high, they overwhelm any initial differences. Therefore it seems essential to know where on this spectrum we are empirically and there seems to be plenty of evidence (including in the experimental component of the paper) that could be better used to parameterize the model.

The model uses 20 individuals. Perhaps this number was chosen to match the experimental data and that may be desirable if the goal is to make quantitative predictions that are most relevant for the experiments but as above, my understanding is that the paper is investigating general theoretical properties. The beauty of simulations is that one is not constrained to realistic sample sizes so why not ramp this up considerably so that the results are less affected by statistical noise. (These simulations should be very fast if coded efficiently.) This is particularly important when the conclusion is a negative one, as in this paper.

In my understanding, the distributions of the naive affinity distributions of different germline alleles are identical between mice (i.e., the value of the s parameter for a given V allele was the same across replicated individuals) and that “epistasis” between a V allele and a D/J gene is represented by a distribution of affinities selected from a truncated normal distribution. Is this correct? Seems to me that there is an odd feature of the model is that when s is large, the realized variance due to epistasis is also larger as few of the samples will be cut off for being negative; the consequence would be that there is more stochasticity in the affinity binding when s is large than when s is small.

I am also confused on the point as to whether the 20 individuals had identical haplotypes (as in the mouse lines used in the experimental setup) or if they shared most of them (as per line 173). A lot of the evidence for germline encoded differences in the adaptive response comes from variation between individuals of different haplotypes.

I think it is inconsistent with the terminology in the evolutionary literature to say that the “neutral” condition requires both no differences in binding affinity AND no differences in mutation rate. It seems completely consistent with a neutral scenario to have variation in the mutation rate across the genome. On this point, the germline-encoded adaptability argument laid out in the introduction suggests that different germline alleles may have different rates of beneficial and deleterious mutations owing to epistasis but in the simulations it is the pure rates of total mutations (which the paper states, line 310, have been shown not to matter) that vary among alleles and not the fitness consequences of mutations. These are conceptually distinct and in my understanding, would be described differently in a mathematical model. I think this affects our interpretation of results (e.g. line 288) does not seem well justified given the set up here.

I am curious as to the precise effect of changes to the number of germinal centers in the model. There are two ways one might consider this: 1) it changes the effective population size of the repertoire; 2) increased competition for the germinal center increases the strength of selection. If I am understanding the model correctly, it is only through the first mechanism that changes in the number of germinal centers matter? What is the rationale behind this choice?

The experiment

I am curious about the choice to use influenza as an experimental antigen, precisely because – as the paper states – it does not naturally infect mice. Most of the reasoning behind germline-encoded specificity is that it is an adaptive response to deal with specific pathogens threats over evolutionary time. It is not clear to me how to extrapolate from these results to cases that are primarily of interest to investigators into germline variation. I would appreciate some extended commentary in the paper as to why this choice was made and how it might impact our ability to generalize the findings.

I do not understand the conclusions of the comparison of the estimated naive V allele frequencies sampled by Greiff et al. 2017? I would like to see an expanded discussion of this in the text; my reasoning is that I am somewhat worried that the correlations between V allele frequencies among individuals that are observed are lower than what we would expect from inbred mouse lines based on Greiff’s study as well as others.

Reviewer #4: Experimental data:

- Can you comment on to what extent you can study your question in an experimental system where influenza is not a common pathogen? Could your conclusions maybe have been different in human where there exists a lot of germline gene variability for IGHV1-69, for example?

- Fig 3b (right plot): why is there so much germline gene usage variation? I would have expected germline gene usage to be more reproducible across mice.

- You state that "Influenza antigens do not strongly select for specific CDR3 sequences"  but the hallmark of antibody sequence is that they can look sequence-dissimilar but are structurally similar? did you test for that?

- Since you don't know the specificity of the antibody sequences studied, it's very difficult to draw strong conclusions from the data. Can you comment on that? In other words, how can you study constriction of antibody repertoire dynamics without knowing the binding landscape (especially given the complex sequence-structure-binding map of antibodies)?

Simulation model

- did you account for the fact that VDJ recombination generates sequences according to a distribution (pgen, generation probability).

- SHM: did you account for the fact that SHM has hotspot motifs and even germline-gene-specific motifs?

**Part III – Minor Issues: Editorial and Data Presentation Modifications**

Reviewer #1: 1) The authors mention the role stochasticity can play. Does the authors model also account for the resultant permissiveness in B cell selection? The monitoring of clonal composition within a primary GC reveals that ‘winner-take-all’ events are rare, and that stochastic factors unrelated to BCR affinity can strongly influence selection, enabling non-homogenizing B cell selection and longer term survival of low affinity B cell clones (PMID 26912368).

2) The authors should include representative flow plots for sorting of the different B cell: the naïve B cells and the GC, memory, and plasma cells that were expanded post-infection.

3) Line 233 (as an example). The authors seem to equate somatic hypermutation (SHM) with affinity maturation. But these are two distinct features. SHM is a repertoire-diversifying process that provides substrate for affinity maturation, but it can easily lead to less fit/lower affinity clones.

4) Line 315 “Most pairs of lineages with the same V allele had no high-frequency mutations in common”. This again seems at odds with some human public antibody responses, which do often contain public mutations that confer gain of function.

5) Do the authors predict a similar relationship for usage of LC V genes?

Reviewer #2: 1. Overlap estimates may benefit from a Morisita’s horn overlap index analysis.

2. In Figure 4 panel B, figures S9,S10,S12,S18 panel B, and S20 panel B. Legibility could be improved by replacing the numbers along the axis with the V Gene names which are overlapping in the current figures.

3. In 4B, plot titles are not clear, are these individual mice names?

Reviewer #3: One overarching comment is that the main takeaway from both the simulations and the experiments is that the early response was indeed constrained by germline-based affinities but that as more and more mutations accumulated the response became less predictable and stereotypical however this early/late discussion is not made in the title, which I feel is misleading.

Reviewer #4: (No Response)

PLOS authors have the option to publish the peer review history of their article (what does this mean?). If published, this will include your full peer review and any attached files.

Reviewer #1: No

Reviewer #2: No

Reviewer #3: No

Reviewer #4: No
---

## [Decision Letter · Decision Letter 1]

7 Aug 2023

Dear Dr. Costa Vieira,

We are pleased to inform you that your manuscript 'Germline-encoded specificities and the predictability of the B cell response' has been provisionally accepted for publication in PLOS Pathogens.

Best regards,

Allan J Zajac

Academic Editor

PLOS Pathogens

Ronald Swanstrom

Section Editor

PLOS Pathogens

Kasturi Haldar

Editor-in-Chief

PLOS Pathogens

orcid.org/0000-0001-5065-158X

Michael Malim

Editor-in-Chief

PLOS Pathogens

orcid.org/0000-0002-7699-2064

The revised report was re-reviewed and the detailed response and updates to the manuscript was appreciated. All of the reviewers were enthusiastic about the study.

Reviewer Comments (if any, and for reference):

Reviewer's Responses to Questions

**Part I - Summary**

Reviewer #1: The authors have addressed the questions raised by this reviewer. Recommend publication.

Reviewer #2: (No Response)

Reviewer #3: I appreciate the researchers engagement with my comments on the previous round. I think the paper is considerably stronger as a result. While certainly not the last word on the subject, this paper is/will be a valuable contribution to our understanding of the importance of genetic variation in the germline IG loci. I have no further comments.

Reviewer #4: I was a reviewer on a previous version of this manuscript. The authors have addressed all of my comments.

**Part II – Major Issues: Key Experiments Required for Acceptance**

Reviewer #1: (No Response)

Reviewer #2: (No Response)

Reviewer #3: (No Response)

Reviewer #4: See above.

**Part III – Minor Issues: Editorial and Data Presentation Modifications**

Reviewer #1: (No Response)

Reviewer #2: (No Response)

Reviewer #3: (No Response)

Reviewer #4: See above.

PLOS authors have the option to publish the peer review history of their article (what does this mean?). If published, this will include your full peer review and any attached files.

Reviewer #1: No

Reviewer #2: No

Reviewer #3: No

Reviewer #4: No

---

## [Editor Report · Acceptance letter]

20 Aug 2023

Dear Dr. Costa Vieira,

We are delighted to inform you that your manuscript, "Germline-encoded specificities and the predictability of the B cell response," has been formally accepted for publication in PLOS Pathogens.

Best regards,

Kasturi Haldar

Editor-in-Chief

PLOS Pathogens

orcid.org/0000-0001-5065-158X

Michael Malim

Editor-in-Chief

PLOS Pathogens

orcid.org/0000-0002-7699-2064